# Administration of Bacteriophages via Nebulization during Mechanical Ventilation: In Vitro Study and Lung Deposition in Macaques

**DOI:** 10.3390/v15030602

**Published:** 2023-02-22

**Authors:** Sandrine Le Guellec, Jeoffrey Pardessus, Elsa Bodier-Montagutelli, Guillaume L’Hostis, Emilie Dalloneau, Damien Piel, Hakim Chouky Samaï, Antoine Guillon, Elvir Mujic, Emmanuelle Guillot-Combe, Stephan Ehrmann, Eric Morello, Jérôme Gabard, Nathalie Heuzé-Vourc’h, Cindy Fevre, Laurent Vecellio

**Affiliations:** 1DTF Aerodrug, Faculté de Médecine, Bâtiment M, 10 Boulevard Tonnellé, F-37032 Tours, France; 2INSERM, Centre d’Etude des Pathologies Respiratoires, CEPR U1100, Faculté de Médecine, 10 Boulevard Tonnellé, F-37032 Tours, France; 3Faculté de Médecine, Université de Tours, 10 Boulevard Tonnellé, F-37032 Tours, France; 4Pherecydes Pharma, 22 Boulevard Benoni Goullin, F-44200 Nantes, France; 5Cynbiose Respiratory, Site MAME, F-37000 Tours, France; 6CHRU TOURS, Service de Médecine Intensive Réanimation, INSERM CIC 1415, CRICS-TriggerSEP F-CRIN Research Network, 2 Boulevard Tonnellé, F-37034 Tours, France; 7DTFmedical, 19 rue de la Presse, F-42003 Saint Etienne, France; 8Direction Générale de l’Armement, Agence de l’Innovation de Défense AID Biologie et Biotechnologies, 60 Boulevard Martial Valin, F-75509 Paris, France; 9UMR1282, Infectiologie et Santé Public, ISP, INRAE, F-37380 Nouzilly, France

**Keywords:** bacteriophages, nebulization, morphotype, in vitro viability, mechanical ventilation, in vivo aerosol deposition, lung deposition prediction

## Abstract

Bacteriophages have been identified as a potential treatment option to treat lung infection in the context of antibiotic resistance. We performed a preclinical study to predict the efficacy of delivery of bacteriophages against *Pseudomonas aeruginosa* (PA) when administered via nebulization during mechanical ventilation (MV). We selected a mix of four anti-PA phages containing two *Podoviridae* and two *Myoviridae,* with a coverage of 87.8% (36/41) on an international PA reference panel. When administered via nebulization, a loss of 0.30–0.65 log of infective phage titers was measured. No difference between jet, ultrasonic and mesh nebulizers was observed in terms of loss of phage viability, but a higher output was measured with the mesh nebulizer. Interestingly, *Myoviridae* are significantly more sensitive to nebulization than *Podoviridae* since their long tail is much more prone to damage. Phage nebulization has been measured as compatible with humidified ventilation. Based on in vitro measurement, the lung deposition prediction of viable phage particles ranges from 6% to 26% of the phages loaded in the nebulizer. Further, 8% to 15% of lung deposition was measured by scintigraphy in three macaques. A phage dose of 1 × 10^9^ PFU/mL nebulized by the mesh nebulizer during MV predicts an efficient dose in the lung against PA, comparable with the dose chosen to define the susceptibility of the strain.

## 1. Introduction

Multi-drug-resistant (MDR) *Pseudomonas aeruginosa* (PA) has been considered by the WHO [1] as critical among the pathogen list for research and development of new antimicrobials. In 2017, the Centers for Disease Control and Prevention estimated that 32,600 cases of hospitalized patients were due to MDR-PA and that 2700 were associated with deaths [2].

Ventilator-associated pneumonia (VAP) is defined as an infection of the lung parenchyma. They are a frequent complication in intensive care units [3,4,5,6], and PA is the most frequent cause among Gram-negative bacteria. VAP is diagnosed when detection of PA is superior to 1 × 10^4^ cfu/mL in sputum or superior to 1 × 10^5^ cfu/mL in bronchoalveolar lavage. PA-associated VAP is difficult to treat due to (i) the natural and acquired antibiotic resistance of PA, (ii) its ability to form biofilm and (iii) the poor antibiotic diffusion into the lungs. These characteristics often lead to an increase in doses in order to maintain clinical efficacy, despite an increased toxicity. To increase the topical drug effect and reduce toxicity, local delivery of the antibiotics directly to the lung has been assessed as well. Goldstein et al. [7] evaluated the efficacy of nebulization of amikacin. It demonstrated heterogeneity in the distribution of the drug, with antimicrobial efficacy in the most affected areas but cellular toxicity in the uninfected ones. To our best knowledge, the direct benefit of the aerosol route compared to the IV route for treating VAP with antibiotics has not been clinically demonstrated [8,9,10,11].

Phage therapy is an attractive approach. Its use could decrease the selection pressure on antibiotic resistance, or it can be seen as a complementary approach to save last-resort antibiotics or, in the worst case, as the ultimate treatment when the antibiotic arsenal is exhausted. The phage ability to self-replicate in the presence of its target theoretically decreases the amount of administration and potentially leads to better compliance. Their specificity limits the impact on the flora that participate in tissue homeostasis. Phages are generally considered safe, via different routes of administration, with a low incidence of adverse events [12]. They may be active in biofilms [13,14]. The use of a phage combination theoretically decreases the frequency of resistance and, in some cases, phages can re-sensitize bacteria to antibiotics [15]. These characteristics make them particularly relevant for the treatment of lung PA infections.

In this context, pulmonary administration of phages, in aerosol form, appears to be an alternative to the systemic route to directly deliver phages into the lung at a sufficient dose. This route promotes the local concentration of phages and their replication in affected areas that are often difficult to access with other treatments. Indeed, phages have the property of replication as long as the pathogen is present, and they disappear following elimination. Thus, the local concentration in infected foci is higher than in uninfected areas.

Animal models of PA lung infection, mainly performed in rodents, using a method of phage instillation directly into the respiratory tract, demonstrated the efficacy of phage therapy [16,17,18].

In human therapy, aerosol replaces instillation, and nebulizers are the preferred devices for generating and delivering phage aerosol during mechanical ventilation (MV). They generate a large amount of aerosol (against pressurized Metered Dose Inhalers), they can be added in the ventilation tubing (not possible with Dry Powder Inhaler), and they require low formulation development in comparison with inhalation powder.

Recently, in a porcine model of Ventilator-Associated Pneumonia caused by PA (VAP-PA), Guillon-Pardessus et al. [19] showed that nebulized phage therapy was associated with a 1.5-Log reduction in bacterial load in lungs compared to untreated porcine, suggesting that inhaled phage therapy was able to treat pneumonia.

Although in vivo data of phage therapy in PA lung infection are encouraging, the efficacy of phage nebulization depends on several parameters. Previous in vitro studies evaluated the effect of nebulizer type on bacteriophage viability [20,21]. Other studies have described that the phage viability upon nebulization also depends on the type of phage [22]. In clinical studies using bacteriophages to treat lung infections, stable and complete eradication of involved bacteria can depend on the phage administration mode, and, often, only one type of bacteriophage is used for the treatment, limiting therapy efficiency [12]. Upon nebulization, phages contained in aerosol droplets can be heated or droplets can be evaporated and, consequently, impact the phage viability. A study from Verreault et al. [23] demonstrated the influence of ambient temperature and humidity on phage viability. In the context of MV, the complexity of the ventilator and circuit also modifies the efficacy of aerosol administration.

The objective of the study was to select and characterize the host spectrum of an anti-PA phage combination and to evaluate phage viability and lung delivery upon nebulization in the context of intubated and ventilated patients. In clinical settings, the dose is a key factor to maximize the chance of clinical success.

In this study, we (i) selected an anti-PA phage combination with a coverage of 87.8% (36/41) on the international PA reference panel [24]; (ii) evaluated the phage nebulization efficacy using an in vitro model to predict lung delivery in intubated and mechanically ventilated patients; (iii) measured the bacteriophage deposition in intubated macaque lungs using a scintigraphy measurement method, a relevant in vivo model to predict lung deposition. Altogether, these data led us to define the phage dose to be evaluated in a clinical trial to treat VAP-PA.

## 2. Materials and Methods

### 2.1. Strains

A collection of 641 *P. aeruginosa* strains was used. These strains were provided by a partner or purchased from a public collection. The international reference PA panel described in De Soiza et al. [24] and the strain panel described in Pirnay et al. [25] were purchased from the BCCM/LMG Bacteria Collection (Ghent, Belgium).

Reference strains used for phage titrations were LMG12228 (belonging to BCCM/LMG Bacteria Collection) for *Myoviridae* and NAR71 (clinal isolate belonging to personal Pherecydes collection) for *Podoviridae*.

### 2.2. Formulation of Phage Anti-Pseudomonas Aeruginosa

#### 2.2.1. Phage Discovery

Phages were isolated from sewage water following the principles of the protocols previously described [26]. Briefly, aliquots of coarsely filtered water were mixed with early-phase bacterial cultures made of several strains of PA. After overnight incubation at 30 °C or 37 °C, culture supernatants were collected and used for double-layer plaque assay, as previously described. Isolated plaques were purified with five successive selection rounds of double-layer plaque assay.

#### 2.2.2. Selection of Virulent Phage

A set of genes (integrase, recombinase and excisionase) associated with lysogeny was searched in the phage genomes annotated by the RAST v2.0 server [27]. Phage lifestyle was also evaluated using BACPHILIP version 0.9.6 [28]. All phages were predicted to be virulent with >90%. In addition, phage genomes were blasted on the GenBank database, and the life cycle of the closest phage homolog was identified. The presence of antibiotic resistance genes and bacterial virulence factor encoding gene in phage genomes was evaluated using CARD [29] and the VF analyzer on the Virulence Factor Database [30], respectively. The absence of prophage, from the production strain, was verified in silico using Phaster [31,32] and the absence of temperate phage expression arising from a prophage lacking the replicase gene was assessed by qPCR in the phage preparation.

Phages were produced as a pharmaceutical drug product by Pherecydes Pharma. This includes phage purity quality controls, such as residual host cell DNA, host cell proteins or endotoxins.

#### 2.2.3. Phage Titration

Phages were titrated by plaque spot assay of serial dilutions, as previously described [33]. Serial dilutions of a phage suspension are spotted on the surface of a Lysogeny Broth 0.75% agar plate containing 0.5 mL of a stationary growth phase culture of a reference strain, which is highly susceptible to the phages.

After an overnight incubation at 37 °C, three phenotypes can be observed (Figure 1a):Plaque-forming units (PFUs) can be seen on the bacterial lawn when the phage is diluted with a total lysis of the spot at high phage concentrations, which correspond to confluent PFUs.A partial lysis of the spot is observed without any PFU at further dilutions.Total absence of lysis, whatever the phage dilution.

The phage titer in PFU/mL can be, thus, defined with the PFU phenotype, as follows:$$\frac{\mathrm{Nb}\;\mathrm{of}\;\mathrm{PFU}\;\mathrm{on}\;\mathrm{the}\;\mathrm{spot}\;\left(*\right)\times \;\mathrm{Dilution}\;\mathrm{factor}\;\mathrm{of}\;\mathrm{the}\;\mathrm{phage}\;\mathrm{on}\;\mathrm{the}\;\mathrm{corresponding}\;\mathrm{spot}}{\mathrm{volume}\;\mathrm{of}\;\mathrm{spotted}\;\mathrm{phage}\;\mathrm{in}\;\mathrm{mL}}$$

(*) preferentially PFU are enumerated on a spot containing 5–20 PFUs.

#### 2.2.4. Phage Efficiency

Phage efficiency was in vitro defined using 41 strains belonging to an international PA reference panel [24], which reflects the organism’s diversity and includes commonly studied clones (such as PAO1, PA14, PAK and LESB58), transmissible strains, sequential CF isolates, strains with specific virulence characteristics and strains that represent serotype, genotype and geographic diversity.

A combination of two scores based on two phage activity testing methods was used: first, the Efficiency Of Plating (EOP) score and, second, the Minimal Inhibitory Concentration (MIC).

To calculate the first score, the plaque spot assay of serial dilutions was used. The titer of the phage was determined on strains of the panel as well as on the highly susceptible reference strain (Figure 1a). The EOP score is defined as follows:$$\mathrm{EOP}=\frac{\mathrm{titer}\;\mathrm{of}\;a\;\mathrm{phage}\;\mathrm{on}\;a\;\mathrm{panel}\;\mathrm{strain}}{\mathrm{titer}\;\mathrm{of}\;\mathrm{the}\;\mathrm{same}\;\mathrm{phage}\;\mathrm{on}\;\mathrm{its}\;\mathrm{reference}\;\mathrm{strain}}$$

It represents the efficacy level of a phage under the panel strains compared to the reference strain. The closer to 1, the more effective it is.

The MIC is determined with the well-known broth microdilution assay used in antimicrobial susceptibility testing. Adapted from standards, one in ten serial dilutions of each phage were performed from 1 × 10^9^ to 1 × 10^3^ PFU/mL and mixed with each strain between 5 × 10^5^ and 1 × 10^6^ CFU/mL in a total volume of 150 µL. As a control, strains were seeded without phages. Plates were incubated overnight at 37 °C, with continuous agitation, and optical density at 600 nm was recorded every hour (Figure 1b). The percentage of growth inhibition (GI%) for each phage dose was calculated as the percentage of decrease in the area under the curve (AUC) of the condition with phage compared to the control with the strain alone until the latter reaches the stationary phase:$$\mathrm{GI}\%=\frac{\mathrm{AUC}\;\mathrm{without}\;\mathrm{phage}\;-\;\mathrm{AUC}\;\mathrm{with}\;\mathrm{phage}}{\mathrm{AUC}\;\mathrm{without}\;\mathrm{phage}\;}\times 100$$

The MIC has been defined as the phage concentration leading to 80% of growth inhibition.

#### 2.2.5. Phage Mix Preparation for Nebulization

Selected phages for nebulization experiments were provided, formulated at 1.0 × 10^10±1^ PFU/mL. A preparation containing a combination of phages (equal titers) belonging to *Podoviridae* and *Myoviridae* morphotypes was extemporaneously prepared in 0.9% NaCl at a final titer of 1.0 × 10^8^ PFU/mL before each nebulization. The combination was named AP-Phage mix for Anti-Pseudomonal bacteriophage mix.

### 2.3. Nebulization Parameters Influencing Phage Viability

Phage integrity can be disturbed during the nebulization process, depending on the type of nebulizer, and the transport of the aerosol, depending on the ventilation settings.

The type of nebulizer, the temperature during nebulization and the humidity rate of the transport circuit were tested for their impacts on phage integrity. For each experiment, the AP-Phage mix was nebulized (all 5 phages together) and the viability measured. The viability is defined as the capacity of the phage to maintain its integrity and, consequently, its activity against strain bacteria.

#### 2.3.1. Nebulizer Devices, Phage Aerosol Collections and Viability Characterizations

Four different types of nebulizers were selected for phage nebulization, according to their technical functionality: a jet nebulizer (Sidestream^®^, 4.0 µm in terms of Volume Mean Diameter (VMD), Philips Respironics, Murrysville, PA, USA); an ultrasonic (US) nebulizer (ATO600, 4.4 µm in terms of VMD, ATO600, DTFmedical, Saint Etienne, France); a vibrating mesh (Aeroneb Solo^®^, 3.7 µm in terms of VMD, Aerogen, Galway, Ireland); and a static mesh nebulizer (prototype, 3.8 µm in terms of VMD, DTFmedical, France).

#### 2.3.2. Phage Aerosol Collections and Viability Characterizations


**Experimental set-up**


Figure 2 presents the in vitro experimental set-up used to study the impact of the type of nebulization process (step one) and allowing for the evaluation of nebulization in the heated/humidified condition (step two). The aerosol is collected by a BioSampler^®^, which was connected to a vacuum pump (Figure 2). The BioSampler^®^ is a bio-collector device (5 mL BioSampler^®^, SKC Inc., Eighty Four, PA, USA) dedicated to gently collecting biological samples from air in a liquid vortex of 150 mM KCL (Potassium Chloride Injection, CDM Lavoisier, Paris, France). The aspiration flow rate was set at 12.5 L/min to obtain a –0.5-bar depression, as recommended by the manufacturer. A protective filter is interposed between BioSampler^®^ and the manometer.

For each experiment, the reservoir of nebulizers was loaded with 2 mL of the AP-Phage mix for vibrating/static mesh, jet nebulizers or with 6 mL for US nebulizer, due to the high residual volume. Loaded volume and residual volume (at the end of nebulization) were controlled by weighting steps.

Time duration of collection was determined by the end of aerosol production by visual inspection. BioSampler^®^ flushing method and KCL sample analysis were conducted as previously described by Guillon-Pardessus et al. [19]. Briefly, BioSampler^®^ is rinsed with 5 mL of KCL and samples were tittered by plaque spot assay of serial dilutions (phage titer in KCL sample). The sodium contained in the suspension was used as a tracer to determine the amount of aerosol droplets deposited in the collection system, and output was calculated by the ratio between the collected aerosol droplets and the nebulizer charge. Briefly, sodium concentration of phage mix or KCL samples collected with the BioSampler^®^ were measured using an iCAP™ inductively coupled plasma–optical emission spectrometer (ICP-OES) (ThermoFisher Scientific, Waltham, MA, USA) with an axial wavelength of detection of 818.3 nm.

The theoretical amounts of phages expected in the BioSampler^®^ (theoretical phage titer in KCL sample) were calculated based on the phage/sodium concentration ratios of the suspension loaded in the nebulizers. Phage viability is expressed as the quotient of “phage titer in KCL sample” divided by “theoretical phage titer in KCL sample”.


**Influence of nebulizer type**


Step one was to connect the nebulizer device alone to the top opening of the BioSampler^®^ (Figure 2—set up with [B]). Nebulization experiments on the AP-Phage mix were repeated three times with each nebulizer at room temperature (22 ± 2 °C).


**Influence of humidity**


Step two aimed to test the influence of humidity on phage viability: a heated humidifier (MR85AFU/MR290, FisherPaykel Healthcare, Auckland, New Zealand) was connected to a ventilator circuit (one way) interposed between nebulizer connector and the BioSampler^®^ (Figure 2—set-up with [A] + [B] + [C]). AP-Phage mix aerosol was delivered with the static-mesh DTF prototype [B] connected to a T piece (DTF, France). Herein, 37 °C-100% humidity was first obtained and stabilized in the circuit before starting nebulization of phage mix through the BioSampler^®^. Six experiments were conducted with a stabilized humidity circuit (37 °C, 100% air humidity) and six others with a dry circuit and humidifier turned off (22 °C, 40% air humidity). BioSampler^®^ flushing and KCL sample analysis were performed as previously described.

#### 2.3.3. Influence of Nebulizer Temperature

During nebulization, the nebulization process can increase the temperature in the liquid reservoir [34]. As mechanical stress, the elevation of liquid temperature during nebulization was considered as a major risk for phage survival. The impact of prolonged heating was studied on phage suspension. A temperature of 45 °C corresponds to the temperature reached by the transducer during the mesh nebulization process. Exposition over 15 min (incubation time) corresponds to the theoretical nebulization time and 80 min a worst case, corresponding to the time to nebulize the maximal filling volume with a low flow-rate mesh nebulizer. Incubation at 65 °C was tested to mimic a high rise in transducer temperature as reservoir fluid empties (last seconds of nebulization). To test the influence of nebulizer temperature on phage viability, phage suspension at 1 × 10^10^ PFU/mL (initial formulation) in Protein LoBind^®^ Tubes (Eppendorf AG, Hamburg, Germany) was exposed at 45 °C in water bath from 15 to 80 min (*n* = 3). Phage titer after heat exposure was compared with that of control kept at 4 °C by plaque spot assay of serial dilutions. Exposure experiments were also performed at 65 °C to explore the phage resistance against high temperature. Phage viabilities were calculated in loss of log PFU based on the difference in the log PFU obtained at 45 °C or at 65 °C with the log PFU at 4 °C (mean log PFU45 °C or 65 °C—mean log PFU4 °C) obtained at each time exposure. Each value of log PFU was a mean of three replicates.

#### 2.3.4. Phage Morphological Analyses before and after Nebulization

Phages were purified by centrifugation on cesium chloride (CsCl) gradient, as previously described by Boulanger et al. [35]. Briefly, the gradient of CsCl (99.9%, Honey Well Research Chemicals, Charlotte, North Carolina, USA) was made in ultracentrifugation tubes by successively layering 0.9% NaCl solutions containing a decreasing density of CsCl (ρ = 1.6, 1.5 and 1.3). The phage suspensions were overlaid on top of the gradient and centrifuged at 120,000× *g* for 3 h at 4 °C in a swinging rotor. After centrifugation, the bluish-white and opalescent band containing intact phages was slowly aspirated. The harvested samples were then laid on top of a second CsCl gradient (ρ = 1.5) and centrifuged at 170,000× *g* for 24 h at 4 °C. To remove CsCl from samples, the purified phages were dialyzed using 20,000 molecular weight cutoff cassettes after the removal of excess air (Slide-A-Lyzer Dialysis Cassette, 0.5–3 mL 20K MWCO, Thermo Scientific, Waltham, MA, USA). Dialysis was performed at 4 °C with 0.9% NaCl, twice for 4 h and then overnight 300–500-times the volume of the samples.

Transmission Electron Microscopy (TEM) analyses were performed on the purified stock suspension, before and after nebulized phages with the static-mesh nebulizer. Formvar/carbon-coated nickel grids were deposited on a drop of samples over five minutes and rinsed three times on a drop of water. The negative staining was then performed in three consecutive contrasting steps using 2% uranyl acetate (Agar Scientific, Stansted, UK) before analysis under the transmission electron microscope (JEOL 1011, Tokyo, Japan).

### 2.4. Aerosol Phage Administration in a Human MV Model

Human MV was mimicked by successively connecting, in this order, a ventilator (Primus ventilator, Dräger, Telford, PA, USA), a ventilation circuit (Adult Ventilation Circuit CVAE160/S, Int’AIR Medical, Bourg-en-Bresse, France) and an endo-tracheal tube (ETT) (7.5 mm inner diameter, Rusch Safety Clear, Teleflex, Wayne, PA, USA) to a lung simulator (Dual Adult TTL Model 5600i, Michigan Instrument, Kentwood, MI, USA) equipped with a protective filter [19]. Nebulization was ensured by the static-mesh nebulizer (DTF medical, Saint Etienne, France), associated with the Combihaler^®^ spacer (Protec’som, Valognes, France) in order to optimize the lung deposition [36] and connected to the inspiratory limb, just before the Y piece in reference to clinical practices in intensive care unit. Phage integrity or aerodynamic parameters of phage aerosol were measured at the ETT outlet.

Figure 3 presents this set-up with three different configurations with [A] or [B] or [C]. Nebulizer was loaded either with the AP-Phage mix in 0.9% NaCl alone, for set-up with [A], or combined with 37 MBq of radioactive tracer ^99m^Tc-DTPA for set-up with [B] or [C].In the set-up with [A], phage viability was determined by interposing the BioSampler^®^ between the ETT and the lung simulator (shunt connection) to collect a fraction of the aerosol and based on phage/Na+ ratio, as previously described. In set-up with [B], output was calculated as the quantity of radioactive aerosol delivered to the lung simulator by interposing an absolute filter between the ETT and the lung simulator. For set-up with [C], the particle size distribution of the aerosol was determined by interposing a cascade impactor (Next-Generation Impactor NGI, Copley Scientific Limited, Nottingham, UK) and measuring the ^99m^Tc activity between the ETT and the lung simulator (shunt connection). In both [A] and [C] set-ups with a shunted connection, an additional airflow, equal to that of the vacuum pump, was added to ensure the shunt connection to avoid ventilation modification. Standard ventilatory parameters were applied in order to ensure reproducibility of measurements within bench evaluations, and not with the objective to perfectly reproduce VAP ventilation setting in the clinics, which represents a limitation: a tidal volume (Vt) = 500 mL, a respiratory rate (RR) = 20 respiratory per minute (RPM), Inspiratory: Expiratory ratio (I:E) = 1; a Positive End Expiratory Pressure (PEEP) = 5 cm H_2_O.

### 2.5. In Vivo Lung Deposition Measurements in Ventilated Non-Human Primate

#### 2.5.1. Animals

Measurement of the aerosol lung deposition was conducted in a non-human primate, the reference model to evaluate, using gamma scintigraphy, the lung targeting and the efficacy of aerosol delivery of a human-inhaled therapy [37,38]. Three healthy female cynomolgus macaques (*Macaca fascicularis*, Vietnam), 3–3.5 kg, obtained from Bioprim^®^ (Baziege, France), were housed under conventional conditions in the animal facility (PST-A, Tours, France) in accordance with the latest European legislation (Directive 2010/63/UE). The experimental protocol was conducted according to NIH guidelines for the care and use of laboratory animals. It was approved by the local ethics committee and recorded under the agreement reference N°11682-201700217166146. Individual body weight was monitored regularly during the study period.

#### 2.5.2. Perfusion Scintigraphy

To determine the absorption of ^99m^Tc γ-rays by macaque tissue, perfusion scintigraphy was first performed for each macaque using albumin macro aggregates labeled with technetium 99m (^99m^Tc-MAA, Pulmocis, Curium Pharma, Paris, France). Following its intravenous injection, ^99m^Tc-MAA is distributed throughout the capillary system of pulmonary region. This step allows for determining the tissue attenuation coefficient of each animal against γ-rays, which would be considered for the calculation of aerosol deposition. It also allows for the drawing of region of interest (ROI), corresponding here to the lungs.

Briefly, macaques were sedated via intramuscular injection of 10 mg/kg ketamine (ketamine 1000, Virbac, Carros, France) and 0.6 mg/mL xylazine (Rompun 2%, Bayer, Leverkusen, Germany). The suspension of ^99m^Tc-MAA was IV injected through the saphenous vein. After ensuring that all ^99m^Tc-MAA was trapped in the pulmonary capillaries, animals were imaged in post-anterior static position using a single-head gamma camera for 120 s. Tissue attenuation coefficient (AC) was calculated by dividing the 120 s scintigraphy lung count by the activity administrated by the IV route. The activity administrated by the IV route was calculated by the difference between the syringe before and after IV administration in terms of radioactivity counted by gamma camera.

#### 2.5.3. Set-Up for Aerosol Delivery under MV and Scintigraphy Gamma Camera Imaging

Static-mesh nebulizer was loaded with 2 mL of AP-Phage mix added with 0.1 mL ^99m^Tc-DTPA. The nebulizer radioactivity charge was measured by counting the radioactivity contained in the nebulizer reservoir using the gamma scintigraphy method. Animals were anesthetized with 10 mg/kg ketamine and 0.6 mg/mL xylazine and maintained under 2% isoflurane. Animals were orotracheally intubated (2.5 mm inner diameter, Rusch Safety Clear, Teleflex, Wayne, PA, USA) and mechanically ventilated in a supine position (Primus ventilator, Dräger). The following ventilator parameters were set: tidal volume, 8 mL/kg; respiratory rate, 32 breath/min; FiO_2_ to maintain a SpO_2_ of 95–100; and a 5 cm H_2_O positive end-expiratory pressure. The nebulizer was connected to the inspiratory limb with its T piece (DTFmedical, Saint Etienne, France) adapted to fit with the neonatal circuit diameter.

Immediately after nebulization, animals were imaged on the gamma camera detector. Lung deposition was delineated using the perfusion scintigraphy ROIs. All radioactive measurements were corrected for tissue attenuation, background activity and radioactive decay of technetium. Amount of deposited aerosol is given for each animal and expressed as a percentage based on the total amount loaded in the nebulizer.

### 2.6. Statistical Analysis

The effect of the type of nebulizer on phage viability was assessed for each morphotype using the Kruskal–Wallis test followed by a post hoc Dunn’s multiple comparisons test. The overall phage viability between morphotypes with the different nebulizers was compared using a paired *t*-test. Among each morphotype, the effect of circuit humidity rate on the loss of titer log was assessed using a non-parametric Mann–Whitney U test. All comparisons were considered statistically significant with a *p*-value < 0.05. GraphPad prism 8.02 Build 263 (GraphPad Software, Inc., San Diego, CA, USA) was used for statistical tests.

## 3. Results

### 3.1. Selection and Evaluation of Phage Mix and Efficacy on Clinical Strains

#### 3.1.1. Phage Discovery

More than 500 phages were isolated from sewage water, mainly collected in Europe but also in America, South Asia and Africa. These phages were purified and their activity was tested on large panels of strains, 641 in total, including 489 non-published strains collected in hospitals on the five continents between 1936 and 2013, as well as 109 strains [25] and 42 strains [24] that were well characterized, previously published and publicly available. The in vitro screening based on plaque assay at a single phage concentration (6–8 Log PFU/mL) led to the selection of nine phages with complementary host ranges or specificity for virulent and antibiotic-resistant clones, such as the pandemic MDR Serotype O12 clone. Among the 641 tested strains, 199 were isolated from lung infections and 111 were classified as Multi-Drug-Resistant strains. Among these sub-groups, 98% (196/199) and 97% (108/111) were susceptible to at least one of the nine phages, respectively.

*P. aeruginosa* virulence factors and antibiotic resistance genes were absent in the genomes of the 9 phages, and they were strictly lytic, with at least 90% confidence.

#### 3.1.2. Phage Selection and Host Range

The last selection was carried out on the international *P. aeruginosa* reference panel [24], which reflects the organism’s diversity (see Section 2). It led to the selection of four phages, PP1450, PP1777, PP1792 and PP1797, that can be manufactured with the same host. The host range was determined for each phage–strain combination. Based on the MIC and the EOP scores, susceptibility classification of the strain to each phage was proposed to use in this study: “S” standing for Susceptible, “I” standing for susceptible at Increased dose and “R” for Resistant.

In the absence of guidelines for phage susceptibility testing, the SIR categories were based on the two scores, EOP and MIC, as described in Figure 4.

According to these categories, the four phages displayed a coverage of 87.8%, meaning that 87.8% of the strains were susceptible (“S” or “I”) to at least one phage and 12.2% were resistant to all four phages. When considering only the “S” category, the coverage was 83%.

The detailed number of strains in each category for each phage revealed a good relation between EOP and MIC (Appendix A). It is noteworthy that the absence of lysis in the plaque assay is, in all cases, associated with resistance (MIC ≥ 1 × 10^9^ PFU/mL) in the broth microdilution assay.

The meaning of the phenotype with lysis of the spot without PFU is questionable. This phenotype is largely associated with MIC > 1 × 10^9^ PFU/mL. In only 1.8% of cases, it is associated with MIC ≤ 1 × 10^7^ PFU/mL (3 phage–strain combinations out of 164). Strains with an EOP < 0.005 are considered weakly susceptible [39], which is confirmed by the fact that only 3% of the phage–strain combinations fall in the category with EOP < 0.005 and MIC ≤ 1 × 10^7^ PFU/mL.

### 3.2. Nebulization Parameters Influencing Phage Viability

Due to technical issues early in the development of the PP1450 manufacturing process, an additional back-up phage, the PP1902, with a similar host spectrum to PP1450 was included in the composition of the phage mix, which was tested in nebulization.

Phages PP1450, P1777 and PP1902 belong to the *Myoviridae* morphotype, characterized by a retractile tail, and PP1792 and PP1797 belong to the *Podoviridae* morphotype, characterized by a short tail.

The impact of the nebulization process on phage viability of both families was investigated using the BioSampler^®^ aerosol collection method (Figure 2) with different nebulizer and humidity rates. The five phages were combined (AP-Phage mix) and nebulized. Then, viable, i.e., infectious, *Myoviridae* and *Podoviridae* were titrated using reference strains specific to each morphotype and then normalized with Na+ quantities produced by nebulization (total collected aerosol).

Phage morphotype comparison revealed that *Myoviridae*, with a mean viability of 22.5%, are significantly more sensitive to the nebulization process than the *Podoviridae* morphotype, which maintained a mean viability of 49.6%, whatever the type of nebulizer (Figure 5).

In addition, large variability in terms of viability after nebulization was observed for both *Myoviridae* and *Podoviridae*, with a viability ranging from 9% to 39% and 34% to 70%, respectively.

Although it was not confirmed by statistics (11/12 tests with *p* > 0.05), the different types of nebulizers seem to lead to different phage viabilities (Figure 5-table). The only statistical difference was obtained with the US nebulizers, which were slightly less deleterious for *Myoviridae* than the jet type (Dunn’s multiple test; *p* < 0.05).

Output was >90% and >60%, in the static prototype and vibrating-mesh nebulizers (in 5 min), respectively, while it was 20% and 25% with the jet type (in 5 min) and the US type (in 4 min), respectively. The high residual volume lost at the end of nebulization process with jet and US—1 mL for jet (2 mL loaded) and 4–5 mL for US (6 mL loaded)—explains these results. Impaction and aerosol loss in the T piece were observed with the vibrating mesh.

The product between the output and the percentage of viability allows one to calculate the quantity of viable phages delivered by the nebulizer. We obtain, respectively, for the static mesh, the vibrating mesh, the ultrasonic and the jet nebulizer for *Myoviridae*: 21%, 15%, 9% and 2.6%, and obtain, respectively, for the static mesh, the vibrating mesh, the ultrasonic and the jet nebulizer for *Podoviridae*: 41%, 36%, 10% and 10%. Altogether, when considering phage viability of each phage family and the output, mesh nebulizers are more efficient in administering these phages.

Based on these viability data, the prototype static-mesh nebulizer was chosen to evaluate the impact of humidity rate. In mechanical ventilation, a heater/humidifier is used to acclimatize and supplies air at clinical conditions of 37 °C/100% humidity, through the ventilated circuit and ETT. These specific climatic conditions were in vitro reproduced (Figure 2). Comparison of the ambient conditions at 22 ± 2 °C, 40% ± 5% and clinical conditions (37 °C, 100%) showed similar titer loss for *Myoviridae* (*p* = 0.9654, *n* = 6) and for *Podoviridae* (*p* = 0.6991, *n* = 6), respectively (Figure 6c).

### 3.3. Phage Morphological Analysis

To understand differences in sensitivity between phage morphotypes, samples of each phage were examined with an electron microscope, before and after the static-mesh nebulization process. Data show the morphologic characteristics of the *Myoviridae* types, with an icosahedral head (62–71 nm) and a long and contractile tail (130 nm–70 nm), and of the *Podoviridae* with icosahedral head (62–71 nm) and a 15 nm short tail (Figure 6a). Before nebulization, all phages were mainly intact. After nebulization, *Podoviridae* had a morphological appearance similar to that before nebulization. On the contrary, images of *Myoviridae* presented more disrupted phages after nebulization than before. Numerous empty and disrupted phages were identified (Figure 6a—left-bottom image). The tails seem to be detached from the phage head (tail alone).

### 3.4. Effect of Temperature during Nebulization Process

At 45 °C, incubations for 15 min do not impact phage viability (Figure 6b). In addition, prolonged incubations, until 60 min or 80 min (as a long inhalation aerosol model), do not have an impact on the *Myoviridae* and on the PP1792 *Podoviridae* (loss < 1 logPFU).

In contrast, exposure to 45 °C for 80 min completely inactivated the PP1797 *Podoviridae* (Figure 6b). Results indicate that *Podoviridae* were more sensitive to prolonged temperature elevation (45 °C) than *Myoviridae*, which are still stable, even incubated at 65 °C for 15 min. After a longer time of exposure, *Myoviridae* began to lose infectiveness by about −1.2 log PFU, corresponding to maximal loss obtained for the PP1777 from 80 min (Figure 6b).

### 3.5. Aerosol Phage Administration in a Human Mechanical Ventilation Model

Standard clinical-assisted ventilation was in vitro reproduced to assess the drug and device proposed for inhaled phage therapy (AP-Phage mix with the selected prototype static-mesh nebulizer). Specific set-up was built to measure phage viability and lung deposition prediction during MV.

At the endo-tracheal tube outlet, the viability of delivered phages was measured at 8.2 ± 2.0% for *Myoviridae* and 35.5 ± 1.0% for *Podoviridae* (Figure 7-line 3A). Viability results are lower than those obtained in standard administration (Figure 5), in particular for *Myoviridae* (8% vs. 23%). *Myoviridae* seems more affected than *Podoviridae* with this supplementary constraint of MV added to the nebulizer.

Based on the nebulization experiment with the AP-Phage mix combined with the radiotracer (^99m^Tc-DTPA), 76.1% of the radioactivity loaded in the nebulizer was delivered at the ETT outlet (Figure 7-line 3B). Thus, 76% may correspond to the fraction of liquid reservoir in a form of aerosol droplets that would be delivered at the trachea, if considering an intubated patient and receiving the inhaled aerosol in the same conditions. This aerosol delivered at the extremity of the ETT has a particle size characterized by an MMAD of 1.5 ± 0.2 µm, 98.0 ± 1.0% of the droplet fraction, with an aerodynamic diameter smaller than 5 µm, and 76.1 ± 8.5% of droplet fraction, with an aerodynamic diameter smaller than 2 µm (Figure 7-line 3C). Based on the similitude of the aerodynamic parameters of the AP-Phage mix aerosol and ^99m^Tc-DTPA aerosol (Appendix A), we validated the ^99m^Tc-DTPA as a tracer of AP-Phage.

Based on this in vitro model, we predict 6% (8% viability at the ETT × 76% of aerosol delivery at the ETT × 98% of particles < 5 µm) of the *Myoviridae* loaded in the nebulizer will be able to reach the lung, including 5% (8% viability at the ETT × 76% of aerosol delivery at the ETT × 78% of particles < 2 µm) reaching the alveoli. For the *Podoviridae*, with their higher resistance to nebulization (35% at the ETT), 26% will be able to reach the lung, including 21% reaching the alveoli.

### 3.6. In Vivo Aerosol Deposition in NHP

During the study period, the animals presented normal clinical signs, and no abnormal variation in their body weight was notified. Animals well tolerated the different procedures (anesthesia, IV injection, intubation and aerosol administration). No adverse effects of prolonged anesthesia on their recovery were notified.

Perfusion images obtained were in accordance with those expected, indicating a good capillary permeability and an intact alveoli-capillary barrier. In terms of the tissue attenuation coefficient (AC), the values obtained for the three NHPs were 0.82, 0.76 and 0.63.

The duration of nebulization of the 2 mL of AP-Phage mix solution combined with ^99m^Tc-DTPA was 7′09 min, 8′16 min and 7′48 min for the three animals. A representative image of lung deposition is illustrated in Figure 8. Aerosol was deposited within the lungs and in the ETT. For 2/3 NHPs, distribution was in favor of right or left lung. The percentage of loaded ^99m^Tc-DTPA that reached the lung ranged from 19.3% to 46.6%, according to the animal, with an average of 34.4% (Figure 8—table). From 5 to 9% of the aerosol was retained in the ETT. For this animal, an aerosol impaction in the T piece occurred during the administration. Based on the similitude of the aerodynamic parameters of the AP-Phage mix aerosol and ^99m^Tc-DTPA aerosol (Appendix A), we can estimate, in this in vivo model, that 8% (23% viability upon nebulization × 34.4% lung deposition) of the *Myoviridae* and 15% (45% viability upon nebulization × 34.4% lung deposition) of the *Podoviridae* loaded in the nebulizer reached the lung, respectively.

## 4. Discussion

Lower respiratory tract infections accounted for more than 1.5 million deaths associated with antibiotic resistance in 2019, making them the most burdensome infectious pathology [40]. The present study focused on ventilator-associated pneumonia caused by *Pseudomonas aeruginosa* (VAP-PA), a multidrug-resistant pathogen and one of the six pathogens recently revealed as associated with the death burden of antibiotic resistance [40]. Patients with VAP-PA had poor survival (<50%), as usually reported in clinical investigations [3,41].

Bacteriophage-inhaled therapy is a potential treatment against VAP-PA, and it is considered as an antibiotherapy alternative or complementary treatment [42] but still reserved for compassionate use, while the regulatory requirements for clinical applications and prescriptions are still in progress. A preclinical porcine model of VAP-PA demonstrated the efficacy of a specific anti-pseudomonal phage mix administered via nebulization [19]. This treatment led to a significant reduction in lung bacterial load. This study and case reports [43] strongly support that inhaled phage therapy could be used for VAP-PA patients, but double-blinded randomized clinical trials are still needed.

In the case of mechanical ventilation (MV), administration of phages by nebulizers is a promising approach for local treatment. Nebulizers transform the phage suspension in an aerosol into fine droplets containing the phages. Phage infectivity, also named viability, can be disturbed during the nebulization process as well as during the transport of the aerosol in the MV circuit.

Sensitivity to the nebulization process has already been studied for different phages but, in general, on phages nebulized alone and based on morphological analyses [21,22]. Our method quantified the infective phages by considering the total aerosol collected and allowed for comparative study between phage family and nebulizer type.

We described first that nebulization influences the phage viability differently according to its morphotype (Figure 4). *Myoviridae* are significantly more damaged (viability of 22.5%) than *Podoviridae* (viability of 49.6%), whatever the nebulizer type.

To better understand the reasons for this viability loss, the structure of the phage particles was evaluated by TEM after nebulization with the static-mesh nebulizer. To ensure a robust interpretation of the TEM images, the phages were ultra-centrifuged before nebulization to remove initial structural damage and, thus, to visualize only those attributable to the nebulization process. As previously observed [44], *Podoviridae* presented an intact structure, while numerous detached tails were present in the *Myoviridae* samples. As confirmed by other studies, *Myoviridae* can easily lose activity by tail detachment [22]. However, the type of stress causing tail breakage is not clearly identified.

The role of temperature was evaluated and revealed that *Myoviridae* are resistant to prolonged and high-temperature exposition (45 °C and 65 °C—until 80 min). The ability to resist at high temperature is already recognized for another phage family, such as siphophages of lactobacillus, due to a genetic predisposition [45]. In contrast, *Podoviridae* seem to be more affected by the 45 °C temperature. These data agreed with [44], who found the same viability loss after a 15 min exposition at 50 °C (−1 log PFU) and at 60 °C (−2 logPFU/mL). In conclusion, it is unlikely that viability loss by tail breakage of *Myoviridae* would be due to high temperature. This conclusion is reinforced by the fact that viability loss was observed in all four tested nebulizers while these nebulizers have different temperature increase properties. The jet nebulizer makes the drug solution cool, while the static-mesh and US nebulizers elevate the liquid temperature. Stress, other than temperature, might be involved.

Between the different nebulizers, *Myoviridae* seem to be more affected with the vibrating mesh and jet than with the US and static mesh (viabilities of 13% and 17% vs. 37% and 23%, respectively) (Figure 4-table). Similarly to our results, Sahota et al. obtained low viabilities for the two *Myoviridae* PELP20 or PELI40 (respectively, 12% and 2% of nominal dose) after nebulization with a jet nebulizer [20]. Phages are submitted to high air injection, atomization and excessive recycling in the jet nebulizer and intensive vibration to allow for droplet ejection in the vibrating mesh. These results can suggest that the tail of *Myoviridae* may be strongly impacted by the mechanical movements induced by the vibrating mesh and jet types. *Podoviridae* were less impacted by these nebulizers, with 52% and 60% viability with the jet and vibrating mesh devices, respectively. Globally, the nebulization process led to a reduction from 0.30 to 0.65 log of infective *Podoviridae* and *Myoviridae* titers, respectively.

High humidity rates of ventilated air, at minima superior to 75% saturated (absolute humidity 33 mgH_2_O/mL), are required in MV to protect the patient from lung injury [46]. In clinical settings, a heating humidifier (connected on the ventilator circuit or respirator) provides and maintains insufflated air at 37 °C/100% relative humidity. Although the humidifier is stopped during the nebulization process, humidity and temperature are still high. We have shown that these parameters have no significant impact on phage viability, whatever the phage family (Figure 5c). This result was also described by Zhang et al. (2021) for a 37 °C biological temperature [44].

Conditions at 37 °C/100% may modify the aerodynamic parameters of the phage aerosol and, thus, droplet size and phage concentration within the droplets due to evaporation or condensation (hygroscopic growth). In comparison with ambient conditions (22 °C/40%), liquid aerosols are mainly affected by evaporation (from nebulizers and metered-dose inhalers) while solid aerosols from dry powder inhalers by growth hygroscopy [47]. Condensation is limited for liquid aerosols thanks to a high concentration of droplets per unit volume, thus keeping a high level of self-humidity in the cloud [47]. Particle size distribution (PSD) of the phage aerosol generated by the nebulizer predicts phage droplet deposition pattern in the respiratory tract and was measured according to the European Pharmacopoeia method (Figure 3 and Appendix A) [48]. The study revealed that aerosol droplets generated with the static-mesh nebulizer were suitable for inhalation and with a pulmonary deposition: 3.4 µm in terms of MMAD, if considering oral inhalation (Appendix A), and 1.5 µm in terms of MMAD when delivered by ETT during MV (Figure 7-line 3C). At the ETT outlet, 98% of the particles had an adapted size for lung penetration (<5 µm) and 78% for alveolar deposition (<2 µm). These performances contribute to the obtention of a diffuse deposition in the whole lung, as revealed in macaque scintigraphy (Figure 8), and are in accordance with standard aerosol requirements for clinical studies by inhalation under mechanical ventilation [49,50].

In addition to humidity, other physical constraints of MV can directly affect the deposition of the inhaled aerosol and potentially the phage viability, such as the ventilator parameters (depending on patients), the accessories volumes and dimensions (long circuit and narrow ETT, high dead space) and the pressure applied into the circuit. Indeed, the phage viability measured during in vitro MV (Figure 7) was lower than the viabilities obtained with a nebulizer alone: 8% for *Myoviridae* and 35% for *Podoviridae* vs. 23% and 45%, respectively, for the static mesh alone.

These viabilities correspond to a slight diminution in terms of log PFU/mL of infective phages, approximately a loss of 0.58–1.2 log PFU/mL (with 2 × 10^8^ PFU/mL for the initial dose loaded in the nebulizer).

The lung deposition of the viable phage was 76%, based on in vitro MV simulating experiments, and between 19% and 47% (average of 34%), based on the intubated macaque study, which is considered as the preclinical model relevant for human aerosol deposition measurements. However, one has to keep in mind that the results were obtained in a healthy lung model, with a ventilator used for anesthesia. Extrapolation to pathological conditions should be performed with caution.

The in vitro lung deposition is higher than macaque lung deposition, mainly due to the difference in circuit and ventilatory parameters. In vitro prediction corresponds to an optimized administration thanks to the use of the inhalation chamber (Combihaler^®^), which concentrates the aerosol during the inhalation phase and stores it during the exhalation phase, optimizing aerosol delivery. It also uses a larger-diameter circuit and higher respiratory volume than the macaque study. Nevertheless, data obtained were in accordance with deposition measurement performed in the preclinical porcine model of VAP-PA, for which the respiratory parameters were closed to adult parameters (tidal volume of 200 mL vs. 500 mL, respectively) [19].

Respiratory parameters of NHPs are close to those of a young human child (3 kg) (a pediatric or neonatal model). Despite this consideration, the drug and device led to phage lung deposition of more than 20% of the initial dose, compatible with an inhaled treatment [49,50].

Data on phage viability upon nebulization and transport into the MV system combined with the lung deposition prediction allow one to predict the % of phages loaded in the nebulizer that will end up in the lung. For *Myoviridae,* according to the in vitro model, 6% of the phages loaded in the nebulizer will reach the lung. For *Podoviridae*, the percentages increase to 26%.

Based on the in vitro data, 83% of the strains are susceptible “S” to at least one phage at a dose ≤1 × 10^7^ PFU/mL. Therefore, to treat VAP-PA, the dose delivered to the lungs must be at least 1 × 10^7^ PFU/mL. Considering the infective phage output delivered by nebulization during MV, we determined in vitro as 6% (for *Myoviridae*) and 26% (for *Podoviridae*); the dose to be loaded in the nebulizer should be calculated as approximatively 2 logPFU/mL higher than the effective dose required in the lung, in order to prepare and supply phage clinical batches. With a concentration of 1 × 10^10±1^ PFU/mL for each phage clinical batch, at least 1 mL of each active phage must be resuspended into a total volume of 5 mL to reach an efficient local dose in the lungs, i.e., a minimum of 1.2 × 10^7^ PFU/mL for *Myoviridae* (6%) and a minimum of 5 × 10^7^ PFU/mL for *Podoviridae* (26%).

Pending regulatory recommendations and international standards, a proposed “SIR” classification is presented here. It was applied to the final selection of the anti-PA phages with the aim of achieving at least 80% coverage on a representative panel. Both the plaque assay and broth assay are two widely recognized methods to characterize phage activity [51,52]. EOP is the gold standard to define phage efficacy, and we propose to calculate the MIC of phages, which is an essential requirement to set the clinical active dose. EOP thresholds have been proposed previously [39], and the MIC threshold was based on a dose that can be delivered locally in clinical settings. However, the “SIR” classification of several categories of EOP-MIC combination can be discussed, such as strains showing lysis without PFU and MIC ≤ 1 × 10^7^ PFU/mL or strains with EOP < 0.005 and MIC ≤ 1 × 10^7^ PFU/mL. The proportion of strains falling into these categories is 1.8% and 3%, respectively. Because the broth microdilution assay has the advantage of providing an in vitro pharmacologically active dose, which is essential for clinical trials, the choice of classification was made based on the MIC data. Furthermore, a change in the classification of these categories does not impact total coverage, since, for these strains, at least one other phage had an MIC less than 1 × 10^7^ PFU/mL and an EOP greater than 0.005.

The “I” category of the proposed SIR classification is also questionable since the active dose cannot be delivered to the lung via nebulization. It should be noted that for these phage–strain combinations, in most cases, the growth inhibition with 1 × 10^7^ PFU/mL, although it does not reach 80%, is not negligible. Moreover, it was demonstrated in vitro that the appearance of phage resistance decreases when different types of phages are administered. Thus, in the clinic, when a strain is “S” to at least one phage, the addition of a phage belonging to category “I” would potentially participate in bacterial control, although its in vitro efficiency is not optimal and would potentially decrease the emergence of phage resistance.

## 5. Conclusions

In summary, in our study:A set of four anti-PA phages was selected, with a coverage of 87.8% on the international PA reference panel, and produced as a drug pharmaceutical product.Five anti-PA phages were tested for nebulization: *Myoviridae* are significantly more sensitive to nebulization than *Podoviridae*; their loss of infectivity is linked to structural damage with tail detachment. This damage appeared to be independent of temperature stress but rather due to shearing stress during the nebulization operation.The nebulization process alone led to a loss of 0.30–0.65 log of infective phage titers.Mesh nebulizers combine a high output delivery with a moderate impact on phage survival.There is no effect of humidification on phage viability during mechanical ventilation.According to the in vitro model of adult mechanical ventilation, the lung deposition prediction of viable phage particles ranges from 6% to 26% of the phages loaded in the nebulizer.According to the intubated NHPs in the in vivo model, the lung deposition prediction ranges from 8% to 15%. In this pediatric model of phages, we observed an asymmetrical deposition in the lungs.A phage dose of 1 × 10^9^ PFU/mL is efficient for nebulization via a mesh nebulizer during MV and it shows a good prediction between in vitro and in vivo (lung) concentrations.

## Figures and Tables

**Figure 1 viruses-15-00602-f001:**
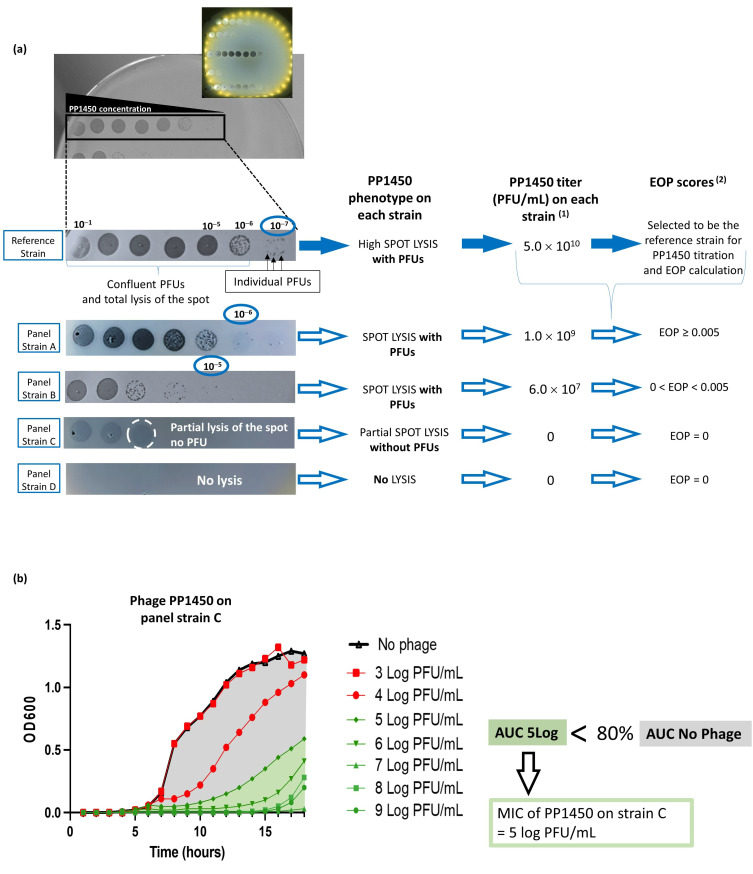
Phage activity testing methods. (**a**) Spot plaque assay method for titration and efficiency of plating (EOP) determination. PFU stands for plaque forming unit or lysis plaque. ^(1)^ titer was calculated as follows: with the reference strain: titer = number of PFU in one spot (25)/[V_phage_ (0.005) × Dilution factor (10^−7^)]; with the strain A: titer = number of PFU in one spot (5)/[V_phage_ (0.005) × Dilution factor (10^−6^)]; with the strain B: titer = number of PFU in one spot (3)/[V_phage_ (0.005) × Dilution factor (10^−5^)]. ^(2)^ EOP score was calculated as follows: score = titer of the phage on the panel strain/titer of the phage on the reference strain. (**b**) Broth microdilution assay for minimal inhibition concentration (MIC) determination: example for strain C, presenting phenotype without PFU, and an EOP score = 0.

**Figure 2 viruses-15-00602-f002:**
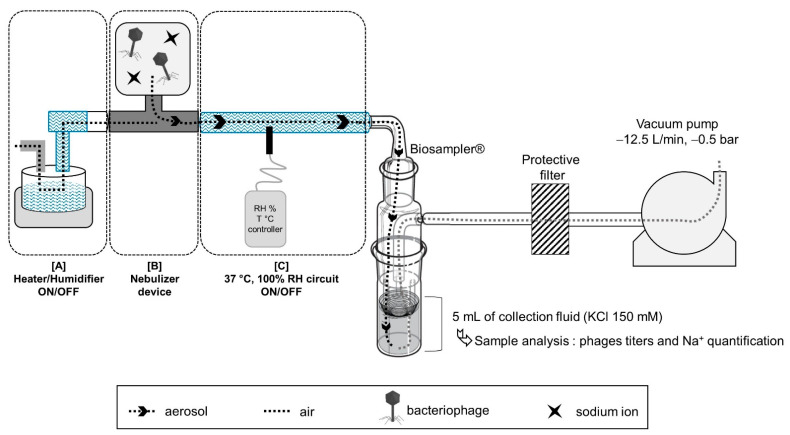
Bench study for in vitro assessment of parameters influencing phage viability: phage morphotypes and humidity. In the center of the figure, the BioSampler^®^ gently aspirates and collects the aerosol containing the bacteriophage and sodium ion from the top opening through a 5 mL vortex fluid of KCl (150 mM), by following −0.5 bar depressurized air from a vacuum pump settled at −12.5 L/min. Step one was to connect the nebulizer device alone [B] to the BioSampler^®^ to assess the impact of different nebulizers on bacteriophage viability. Step 2 [A] + [B] + [C] was to aerosolize bacteriophages with a specific nebulizer device connected to a heated and humidified circuit. For the two steps, after aerosol collection, samples were analyzed to quantify bacteriophage titers using a soft-agar overlay method (infective phage titration) and sodium ions using ICP-MS (output rate of aerosol collection).

**Figure 3 viruses-15-00602-f003:**
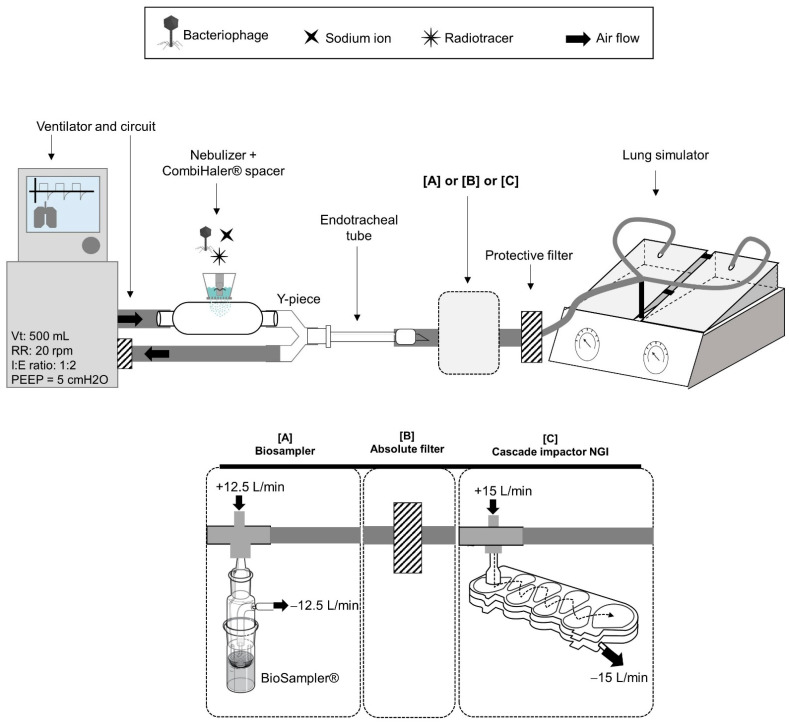
Bench study to assess the aerodynamic parameters of phage aerosol produced by nebulization during human mechanical ventilation. Mechanical ventilation was mimicked by connecting a ventilator, a ventilation circuit and an endotracheal tube to a lung simulator equipped with a protective filter. Nebulization of phages was ensured by the static-mesh nebulizer, associated with a Combihaler^®^ spacer and connected to the inspiratory lamb, just before the Y piece in reference to intensive care unit clinical practice. Three aerodynamic parameters of phage aerosol were measured at the endotracheal tube outlet: [A] phage viability, by inter-imposing the Biosampler^®^ (shunt connection), [B] the output as the quantity of aerosol delivered to the lung simulator, by using an absolute filter, [C] the particle size distribution of the aerosol, by inter-imposing a cascade impactor (shunt connection). In both [A] and [C] set-ups with a shunted connection, an additional airflow, equal to that of the vacuum pump was added to ensure the shunt connection to avoid ventilation modification. Nebulizer was loaded either with the AP-Phage mix in 0.9% NaCl alone, for set-up with [A], or combined with 37 MBq of radioactive tracer ^99m^Tc-DTPA for set-up with [B] or [C]. Standard ventilatory parameters were applied. Vt: tidal volume (500 mL), RR: respiratory rate (20 respiratory per minute), I:E: Inspiratory: Expiratory ratio (1:2), PEEP: Positive End Expiratory Pressure (5 cm H_2_O).

**Figure 4 viruses-15-00602-f004:**
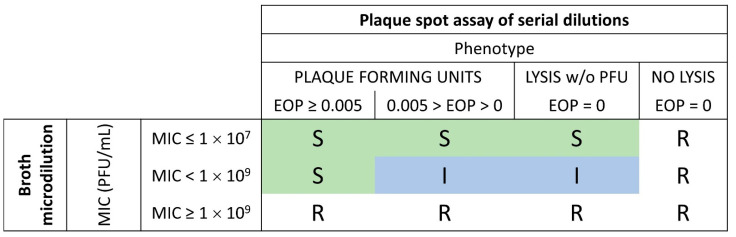
SIR classification based on the MIC and EOP scores: Combining of the EOP and MIC results obtained from phage activity testing methods (Figure 1) and interpretation of the strain susceptibility. The strain was considered as resistant “R” when no lysis was observed on the spot plaque assay or when the MIC was above 1 × 10^9^ PFU/mL. The strain was defined as susceptible “S” as long as the MIC was below 1 × 10^7^ PFU/mL, and when the MIC was between 1 × 10^7^ and 1 × 10^9^ PFU/mL with an EOP above 0.005. The strain was defined as susceptible at increased exposure “I” when MIC was between 1 × 10^7^ and 1 × 10^9^ PFU/mL and an EOP above 0 still 0.005 or when a partial lysis of the spot without PFU was observed on the spot plaque assay.

**Figure 5 viruses-15-00602-f005:**
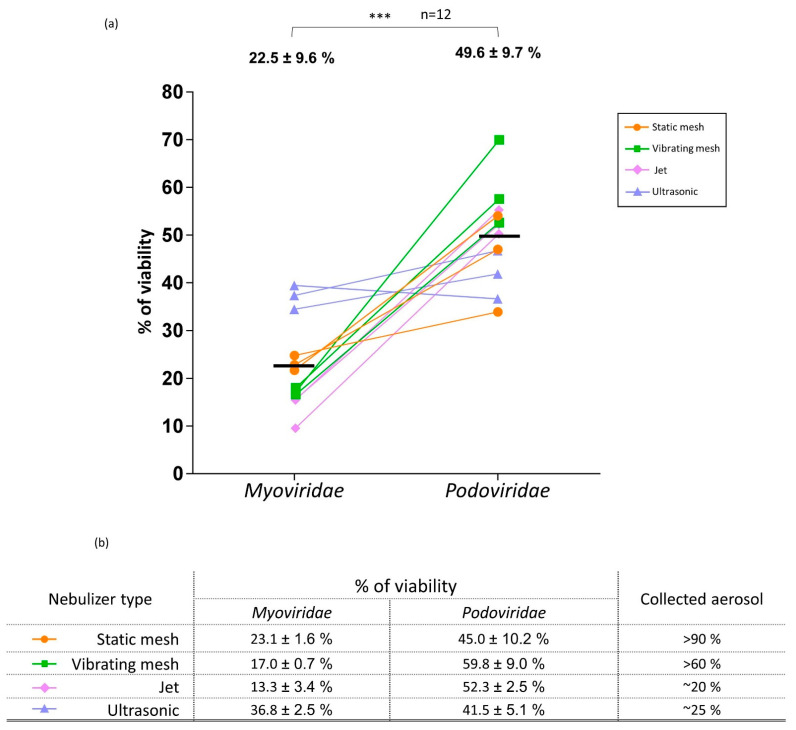
Nebulization effect on phage viability depending on the morphotype. (**a**) Percentages of viability of Myoviridae and Podoviridae after nebulization with different types of nebulizers. Each type of nebulizer is identified by a color. Mean% ± SD of viabilities obtained for each morphotype is indicated at the top of the dot plot and *** = *p* < 0.001 (*t*-test, *n* = 12). Horizontal bar indicated the mean value (*n* = 12). (**b**) Mean% ± SD of viability of Myoviridae and Podoviridae obtained for each type of nebulizer. Collected aerosol corresponds to the quantity of aerosol recovered in the BioSampler^®^ using Na+ as a tracer and expressed in % of the initial dose.

**Figure 6 viruses-15-00602-f006:**
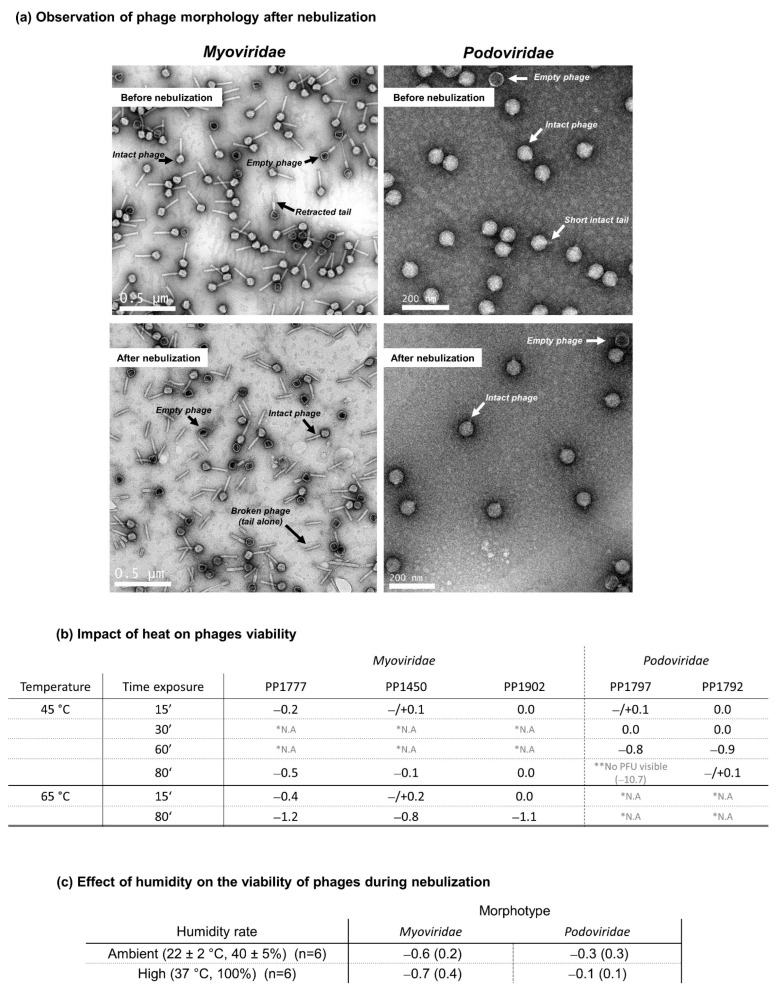
Impact of nebulization on phage morphology and impacts of heat and humidity during nebulization on phage viability. (**a**) Observation of phage morphology after nebulization: Representative Transmission Electron Micrographs of *Myoviridae* and *Podoviridae* before and after nebulization with the prototype static-mesh nebulizer. The white scale lines correspond to 0.5 µm for *Myoviridae* and 200 nm for *Podoviridae*. The white or black arrows show and name the morphological structures observed. Intact phage is considered as a phage with its head, its genome and its tail. (**b**) Impact of heat on phage viability: For each time exposure, the phage viability was expressed as a loss of log PFU, based on the difference between the log PFU measured at 45 °C or at 65 °C with the log PFU measured at 4 °C (mean log PFU45 °C or 65 °C—mean log PFU4 °C). Each value of log PFU was a mean of three replicates. * N.A., for Not Available, means that the time exposure was not tested; ** no PFU visible means that any lysis spot was visible. (**c**) Impact of humidity on the viability of phages during nebulization: loss of log pfu (mean ± SD) after nebulization under ambient humidity (22 °C, 40%) or high humidity rate (37 °C, 100%).

**Figure 7 viruses-15-00602-f007:**
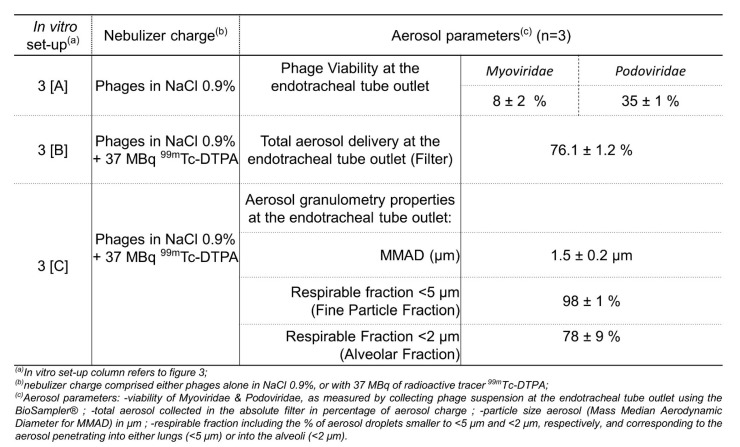
Aerodynamic properties of phage aerosol produced by the static-mesh prototype nebulizer and delivered during adult human mechanical ventilation.

**Figure 8 viruses-15-00602-f008:**
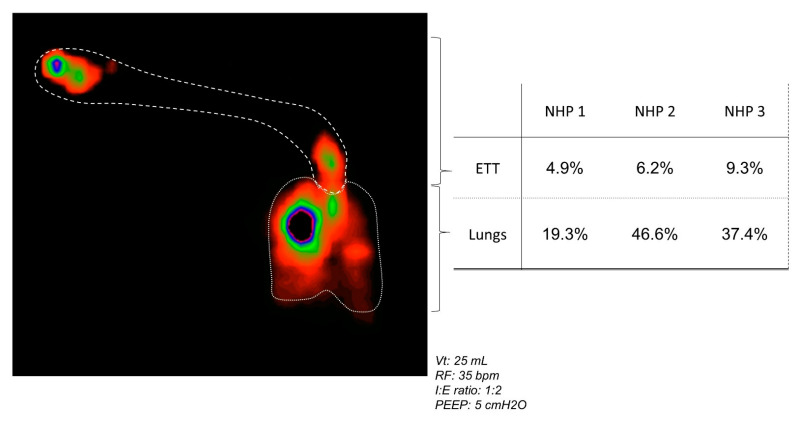
In vivo aerosol lung deposition. (**left**) Representative image of aerosol (AP-Phages + Tc99m-DTPA) deposition in intubated and mechanically ventilated non-human primates. The respiratory parameters were: 25 mL for tidal volume (Vt), 35 breaths per minute (bpm) for respiratory frequency (RF); an inspiratory: expiratory ratio of 1:2; 5 cm H_2_O for positive end expiratory pressure (PEEP). (**right**) Quantity of aerosol deposited in the lungs and endotracheal tube (ETT), and expressed as the percentage of the radiotracer quantity loaded in the nebulizer and given to each animal.

## Data Availability

Not applicable.

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
