# Peer review of "Administration of Bacteriophages via Nebulization during Mechanical Ventilation: In Vitro Study and Lung Deposition in Macaques"

_viruses, 2023, doi:10.3390/v15030602_

Round 1

Reviewer 1 Report

Dear Sir or Madam,

I read the work of Sandrine Le Guellec et al. with a great interest. This is a valuable study on how phages can be administrated with a help of nebulization. The presented findings will help in future to optimized phage therapy for patients with pneumonia and related conditions. I am impressed with extensive experimental data and I had very good reading experience. I appreciate all the work that was invested in this manuscript. I have no major comments. Below I listed usually minor topics, perhaps some may help authors improve further this advanced version of manuscript.

1-3 - It seems that there are some words missing in title like “Administration of (…) during (…); (…)”.

34 – “and linked to structural damage” -> “since their long tail is much more prone to damage”

109 – what does coverage mean? If it is a host range, perhaps give n of tested strains?

127 can phage genomes be publically accessed?

200 how reference panel of strains is defined, how well they represent different linages or phenotypes, geog. origin, clinical sources, etc of studied pathogen (some info is provided in502-504, perhaps a give bit more detail in m&M). any reference culture collections strains were included? Future comparison between different studies will profit from standard strain panel. Perhaps You may consider submitting few critical strains to CCUG or DSMZ I the future, it will increase futher importance of Your work

227 MIC was defined as 80% of inhibition. Why only calculated for 20h. Curves from Fig. 1b clearly indicate strong upwards trend why this information from later time point is neglected?

505 and few other spots P. aeruginosa – please use cursive

776-798 in future studieds it would be interesting to collect low invasive biopsies and perform tissue histology and phage counting

996 – 997 What would be a coverage of Podoviruses alone compared to full cocktail?

1010 “9” goes in superscript

Reviewer 2 Report

In this manuscript as per the title the authors try to simulate administration of bacteriophages in vivo and in vitro via nebulization in mechanical ventilation.

While the topic is of great importance, the manuscript itself is dealing with a broader goals as is described on the 2nd paragraph of page 3.

In my view the manuscript is falling short due to the try to cover such a broad information and should be divided into probably 2 separate manuscripts:

1.       Dealing with Aim 1  (results section 3.1 which is described in extremely limited way).

2.       The rest of the manuscript dealing with nebulization of phages.

Major comments:

1.       Please describe in detail the source of PA isolates. The authors gives quotes to a collection (ref 33) that none of the authors was involved in obtaining. How the access to collection was done? And please also give full credits to this source.

2.       Abstract and line 109: In the abstract, the authors claimed 87.8% coverage which is not consistent with L109 -  95%. Furthermore, it is not clear what was the PA collection upon which these numbers were determined? What was its source? How many strains?

3.       Please describe in detail the coverage experiments of the 9 phages of the full collection of PA what is the difference between the 641 PA isolates and the 199 PA isolates? What is the source of the 199 PA isolates?

4.       In the abstracts it is mentioned 5 isolates with 87% coverage in the results 9 isolates with 97%?  And then only 4 were selected? What are the correct numbers?

5.       There are no EUCAST criteria for phages! And there is no good analogy between phages and antibiotics. With that in mind the table presented in figure 4 is probably wrong. Lysis without PFU can not be defined as S it should be R similarly the EOP<0.005 should be defined as R as well.  The authors need to refer to few publication on this topic – (Lancet Microbe 2021 Oct;2(10):e555-e563.  doi: 10.1016/S2666-5247(21)00127-0. ; J Appl Lab Med . 2022 Oct 29;7(6):1468-1475. doi: 10.1093/jalm/jfac051. And Antimicrob Agents Chemother . 2022 Mar 15;66(3):e0207121. doi: 10.1128/AAC.02071-21. Taking that into account the full panel of susceptibility results need to be reported and disclosed.

6.       Fig 1 and lines 213-235: Does MIC of a multiplying agent has a meaning? Unlike antibiotics, phages may reach the same titer if one waits enough. Thus, for such a calculation, the time range of the experiment and the ability of the phage to kill stationary-phase bacteria need to be taken into consideration. The length of the experiment is not mentioned by the author and is varied between phages. Therefore, it would be very difficult to use it. For instance, in Fig1 the authors waited 17 hours, but it seems that if they waited more, all curves would come to the same point.

7.       Moreover, the MiC determination depends on the initial titer of the phages, thus EOP and MIC, as shown here, are not independent scores!

8.       Please disclose what are the reference strains used for the nebulization studies and how one can obtain those.

9.       It is impossible to understand figure 5. Please revise.

10.   Phage morphology is part of the phage characterization

11.   If I got it right – phages were not labeled with TC99 but rather dissolve in a solution contain TC99 – if that is correct than the whole experiment described in figure 8 and in section 3.6 is wrong. The main reason that is wrong is that the distribution of the TC99 has no relevance to the phages which can be in a different location due to different properties of the phages and their attachment sites..

12.   Discussion is too long – breaking article to 2 manuscripts may facilitate shorter discussion as well.

13.   The ventilator used here (section 2.3) is one that is aimed for anesthesia – and does not reflect similar flows as used in mechanical ventilation for intubated patients due to lung disease. Please at least put that in the limitations section.

14.   Model described in figure 2 – si misleading model since the flow is made by suction and not by "push". That can affect severely the results of phage survival.

15.   Section 2.1.3  - EOP should described in LOG reduction. (3 log reduction should be defined as resistance.

16.   Please do not use EUCAST or CLSI – non has any references for phage susceptibility – please refer to references mentioned in item 4 above. Lines 30-31: Was the reduction observed when using the mix, or was each phage tested individually? If as a mix, how do you know the contribution of each phage to the reduction?

17.   2.1.1. Phage discovery:

a.       Which PA strain was used for the screen?

b.       The author claim that “Although with the phage discovery process including multiple selection and purification rounds it is unlikely to end up with temperate phages” why? If the host strain has a prophage, it doesn’t matter how many purification rounds will be performed, the prophage will be induced again and again.

18.   Line 130: How did the authors reach the calculation of a 10% reduction in confidence of the virulence?

19.   How did the author concentrate the phages to 10^11 PFU/ml? on which strain was it measured? Did they perform other steps of cleaning?

20.   L506 How the presence of PA virulence was tested? And how the 90% confidence was calculated?

Minor Comments:

1.       Pseudomonas need to be in italics throughout the paper

2.       Line 30: 87.8% of how many PA strains? Which collection was tested in order to select these phages for the experiment?

3.       Line 70: Phages are not always efficient against biofilm

4.       In-vivo and in-vitro should be in italics

5.       Ref 27 directs two links, which one is the right one?

Reviewer 3 Report

Overall a helpful and needed study as phage therapy is entering clinical trials without a lot of pre clinical assessment of delivery and PK/PD.  Ventilator associated pneumonia was not high on my list of infections to treat with phage therapy but it does make sense, and it makes sense to study the ventilator circuit’s effect on delivery if this an indication for which phage could be applied. 

  1. There are some word choice and syntax in some areas which read awkward to me - particularly the repeated use of “sensible” in describing phage in the discussion. 
  2. Abstract states the study was performed “to predict efficacy of bacteriophages against Pa when administered by MV”, I would add predicted efficacy of delivery (as you are not assessing the killing of Pa in either the in vitro or in vivo model. 
  3. Methods section - there are almost no references. Many of the methods have references in the discussion but this belongs in the methods too.  Specifically the two models used and the assessments of perfusion and deposition.  If there is no precedent or reference then I would appreciate a line or two about how the methods was developed or justified.  I do really like the diagrams/figures yo have made to depict your experimental set up.  
  4. In the conclusions the authors state “produced as a pharmaceutical drug product” but that term is not used elsewhere.  What does that mean?  Was this produced with FDA or EDU standards like GMP? There was also no discussion how he phages were propagated and purified which should be included. To be considered for clinical use also an LPS or endotoxin content should be reported as well.  
  5. Page 2 line 50-52 - think supposed to be 10^4 and 10^5 not 104 and 105?
  6.   Page 2 line 83-84 - this sentence is awkward and I don’t know what the authors are trying to communicate
  7. Page 3 line 142 and 143 - seem to be incomplete sentences?
  8. Page 6 line 124 - is that a title that wasn’t formatted correctly or an incomplete sentence?
  9. “SIR” abbreviation was used a few times and I couldn’t find what it stood for
  10. Page 19 - line 958 - what company are you referring to?  Please provide reference and more info (I agree that is close to the doses I’ve seen being used in case reports/clinical trials, but more info needs to be provided)
  11. The last paragraph of the discussion seems to come out of nowhere - like it, it think ti is an interesting point - maybe adding some of an intro the concept that this is a different way of looking at it in your methods or results section would be helpful to set the scene.  And then I’d try to somehow connect the rest of your discussion to it, make it a better transition.  
  12. The last thought I have is that it would have been really cool to do some PK or PD measurements in your in vivo model.  I’m not at all saying you need to do that to publish this paper, but it was sort of a missed opportunity.  Those of us working on designing phage therapy trials would love to have that data.  While each phage is likely different and an infected host vs not infected probably behaves differently, I think ti would still be informative.  So next experiments you take on like these, think about collecting PK and PD data!  

Round 2

Reviewer 2 Report

n/a

Author Response

Here is the final version of the manuscript.